# Deep Learning-Based Motion Style Transfer Tools, Techniques and Future Challenges

**DOI:** 10.3390/s23052597

**Published:** 2023-02-26

**Authors:** Syed Muhammad Abrar Akber, Sadia Nishat Kazmi, Syed Muhammad Mohsin, Agnieszka Szczęsna

**Affiliations:** 1Department of Computer Graphics, Vision and Digital Systems, Faculty of Automatic Control, Electronics and Computer Science, Silesian University of Technology, 44-100 Gliwice, Poland; 2Faculty of Automatic Control, Electronics and Computer Science, Silesian University of Technology, 44-100 Gliwice, Poland; 3Department of Computer Science, COMSATS University Islamabad, Islamabad 45550, Pakistan; 4College of Intellectual Novitiates (COIN), Virtual University of Pakistan, Lahore 55150, Pakistan

**Keywords:** deep learning, deep neural networks, human motions, motions style transfer, motion datasets

## Abstract

In the fourth industrial revolution, the scale of execution for interactive applications increased substantially. These interactive and animated applications are human-centric, and the representation of human motion is unavoidable, making the representation of human motions ubiquitous. Animators strive to computationally process human motion in a way that the motions appear realistic in animated applications. Motion style transfer is an attractive technique that is widely used to create realistic motions in near real-time. motion style transfer approach employs existing captured motion data to generate realistic samples automatically and updates the motion data accordingly. This approach eliminates the need for handcrafted motions from scratch for every frame. The popularity of deep learning (DL) algorithms reshapes motion style transfer approaches, as such algorithms can predict subsequent motion styles. The majority of motion style transfer approaches use different variants of deep neural networks (DNNs) to accomplish motion style transfer approaches. This paper provides a comprehensive comparative analysis of existing state-of-the-art DL-based motion style transfer approaches. The enabling technologies that facilitate motion style transfer approaches are briefly presented in this paper. When employing DL-based methods for motion style transfer, the selection of the training dataset plays a key role in the performance. By anticipating this vital aspect, this paper provides a detailed summary of existing well-known motion datasets. As an outcome of the extensive overview of the domain, this paper highlights the contemporary challenges faced by motion style transfer approaches.

## 1. Introduction

In the fourth industrial revolution, the use of interactive applications has increased substantially. These applications play a vital role in several domains, such as video games, robotics, virtual reality (VR), and animated films, among others. The proliferation of portable devices and the advent of 5G communication technologies further accelerate the usage of interactive applications. The interactive and animated applications extensively use human motion modeling. The movement of human-like characters is a core aspect of these applications, and it is highly desirable that the movements of characters look natural. The motion styles lead to the addition of realism to the character movements in these applications.

One of the key challenges in interactive applications is bringing life to animated characters and making them behave naturally. The addition of styles makes the motion more meaningful, and significant research efforts are being carried out in this aspect. The definition of the concept of motion style is not precise, and several researchers describe it differently [1]. The motion of a character is determined by its actions, whereas the style is determined by how those actions are performed. For instance, a character may perform a walk from one location to another in a variety of ways, such as fast, slow, happy, sad, etc. The action itself, i.e., walking, is referred to as motion, and the manner of performing the action, i.e., fast or slow, is known as style.

Defining the concept of *style* is not an easy task due to the lack of consensus. We summarize the existing interpretations and illustrate this term in the graphics industry. Generally, the style of human body motion is referred to as an essential component for executing the complete motion of a character. Another option is to decompose motion into primary and secondary themes and treat actions as primary motions and the style corresponding to the motion as a secondary theme [1]. Massih et al. [2] describe style as a function of age, gender, and health.

Several researchers [3,4] mention style as part of the motion and consider it an *add-on* to motion. Rose et al. [3] describe style as an *adverb* and parameterized motion as a verb. For example, *walk* is a verb; *quick*, *slow*, and *happy* are some of its adverbs. Another way to define style is to consider style as a variation of motion [5]. Other researchers refer to style as a person-dependent feature that may depend on the age, gender, or emotion of the individual [1,2]. In conclusion, style is associated with the personality of the human and reflects the spatiotemporal variations of a movement that supports adding value [1,6].

Styles in human motion may be perceived as a large collection of motion parameters that depict the mood of a character. Style variations have the ability to portray a character’s emotions, such as happiness or sadness, through their posture and walking style. Generally, one can easily tell whether a person is angry or not from the way they walk. Such style predictions are crucial in games and movies based on expressive and realistic character animation. However, generating all the required motions with a diverse set of styles is impractical. The generation of heterogeneous stylized motion is an active area of interest for researchers.

To perform human body motion and style that is analogous to natural motion and to deploy the mapping in animations, researchers use several approaches. The most straightforward approach is manually creating motion data from scratch by some 3D animators; however, such a manual method may be extremely time-consuming and inefficient. Another approach is to use motion capture technologies and map the human body motion data in 3D. However, such motion-capturing technologies may consume more resources since these approaches need large numbers of motion and style combinations and, thus, may become costly [7].

Furthermore, several data-driven techniques [7] were published; they were trained via regression by employing a large motion database and matching motion sequences and styles [8]. Obtaining a comprehensive training dataset for heterogeneous motion sequences and styles is indeed a challenging task. Additionally, several motion-matching methods have been proposed by researchers, based on the optimization theory, which can update the real-time performance of motion matching [9]. However, retraining for new transformations and keeping the motion dataset updated pose challenges for such approaches.

Interactive animation applications require processing a large amount of motion data to ensure that the animated characters look realistic. Bringing animated characters to life is not an easy task, as thousands of motion attributes need to be triggered in the desired sequence for the movement to look realistic. As the demand for animation applications increases, the need for more scalable methods to generate realistic animations with a higher degree of responsiveness also increases. An alternative option is to update the styles of current motion into one extracted from another. This approach is often referred to as motion style transfer in the literature.

Motion style transfer is an effective solution for mapping input motion into various styles. Generally, motion style transfer approaches employ existing motion data, automatically generate realistic samples, and update motion data [10,11]. Transferring motion styles across various characters has the potential to save time for animators as it avoids the need to generate different motion variations for each animation frame. A single set of motions can be adopted for different characters automatically.

The majority of existing works focus on data-driven motion style transfer techniques [1,7,8], which require motion dataset examples for neutral and derived styles. These motion datasets need to be aligned for the generation and transfer of associated poses. The efficiency and performance of such data-driven schemes depend on the datasets and, therefore, the datasets need to be comprehensive enough to cope with all potential motions. However, creating such a dataset is a challenging task.

Motion style transfer techniques face two fundamental challenges. Firstly, they require paired and registered motion data [11]. To build such data, each character needs to perform specific actions in different styles, e.g., the character needs to perform a walk in several different styles and all such styles need to be paired and registered together. This is a tedious and difficult task to achieve. Secondly, since it is challenging to have a comprehensive dataset consisting of all possible style variations, motion style transfer techniques need to extract and transfer styles from fewer motion samples.

With the increase in the popularity of DL based algorithms, these are being successfully applied in various domains [12,13,14]. In the animation industry, animators have shown great interest in applying DL-based algorithms to animation. Deep neural networks (DNNs) and its variants, such as GNN [15,16,17], CNN [18,19,20], etc., are popular in interactive applications [21]. The reason for adopting neural network-based models for animation is that they use less memory, have faster runtime, and are more scalable. Several researchers [1,7,11] have shown that neural networks can be applied to generate natural motion in various interactive applications.

This paper summarizes the existing state-of-the-art DL-based motion style transfer approaches. We present the contributions as follows:We present a brief overview of the existing state-of-the-art motion style transfer techniques. In the overview, we focus on their architecture, contributions, datasets, and implementation details.We present a comprehensive review of the key enabling technologies for motion style transfer techniques.We present a review of various motion capture datasets. The review of the dataset focuses on the size of the dataset, motions included, frame rate, availability, and several other parameters.As an outcome of the comprehensive review of motion style transfer techniques, existing challenges that preclude the popularity of motion style transfer techniques, are also presented.

The rest of the paper is organized as follows. Section 2 describes the existing popular state-of-the-art motion style transfer techniques. Section 3 briefly describes the tools and technologies for motion style transfer. Section 4 presents well-known datasets for skeleton motion; research challenges are focused on in Section 5. Lastly, Section 6 concludes the paper.

## 2. State-of-the-Art Literature on Motion Style Transfer

Over the years, several techniques have been proposed to generate stylized human motions. Motion style transfer approaches, in which the style of one animation clip is transferred to another while preserving the contents of the latter, have gained considerable popularity over the last decade [22]. Several data-driven and AI-based approaches [23,24] have accomplished motion style transfer for animation. However, the lack of paired and labeled motion data precludes the popularity of such approaches. To address these challenges, DL-based motion style transfer approaches have achieved remarkable success in this domain. This section briefly describes the existing state-of-the-art motion style transfer techniques.

In the animation domain, motion style is a key aspect. Motion style describes how an action is likely to be executed. For human motion, the motion style can portray the mood, personality, identity, and feelings of the human. By revealing information about a human’s internal state, the motion style can provide insight into the character itself. Motion style transfer has the potential to substantially reduce the workload by enabling users to reuse already captured motions. It allows modification of existing motion data and generates new styles without compromising the original contents.

The authors of [1] presented a motion transfer scheme based on a meta-network, which transfers the style of input motion to the output motion while retaining the behavior of primitive motion. The goal of Pan et al. [1] was to establish a generative model that could extract and transfer human motion styles across different frames. The model is divided into three sub-tasks that use fully connected layers to establish the linkage between extracted styles and the input motion. This three-layered fully connected network is known as the meta-network. Establishing a mapping between extracted styles and the input motion is a key operation that enables obtaining the matching transformation for every character in a diverse motion style. Afterward, to transfer the motion style, a convolutional feed-forward network was trained for the transformation function.

The model by Pan et al. [1] does not require data labeling and employs convolutional layers in an unsupervised fashion. The meta-network obtains style attributes extracted from the motion style at run time and establishes a link transformation network for motion style transfer. The calculation of style attributes of motion is performed in conjunction with the loss network and output values. A meta network-based motion style transfer from motion-captured datasets is shown in Figure 1.

Four different publicly available motion datasets were used by Pan et al. [1] for the implementation of their proposed motion transfer model and to generate results. The validation of the results was carried out by comparing them with existing work. The comparison shows that Pan et al.’s model outperforms the existing models.

Holden et al. [7] proposed a neural network method for motion style transfer that consists of two components, i.e., a convolutional auto-encoder and a feed-forward convolutional network. The former is used as a loss network, while the latter is responsible for the style transformations. The convolutional auto-encoder models human motion and calculates the loss during style transfer. Figure 2 presents an overview of Holden et al.’s [7] method for motion style transfer.

The authors of [7] used the motion dataset created by combining several freely available datasets and adding their own capture motions. The data are converted to the 3D joint position format; foot contact labels are detected by observing the toe and heel of the character. The joint position representation is very effective at computing the Euclidean distance between two poses to determine the similarity of the poses. The data representations are fed in a fixed-size window for the training. The training phase took 6 h on an NVIDIA GeForce GTX 660 GPU.

Holden et al. [7] generated results via 10 different locomotion and style types. The results were evaluated by comparing them with the existing style-transfer method and using the runtime as the baseline for the evaluation. The evaluation concluded that Holden et al.’s [7] model produces a better runtime.

Smith et al. [8] propose a method that uses neural networks for motion style transfer and generates stylized motion. To implement this, Smith et al. prepare a set of networks that take diverse motion sequences as input and produce the same sequence with a specific style as output. As long as the joint positions can be determined, any exciting motion can be effectively streamed into the networks as input for transfer. Thus, there is no need for retargeting or input action labels.

The proposed method primarily divides the different components of the task for motion style transfer and trains each task separately. The style transfer tasks are divided into three sub-tasks: (i) a pose network, (ii) a timing network, and (iii) a foot contact network. All of these sub-tasks are separately trained to accomplish their respective predictions. Specifically, pose networks and timing networks are used to predict the spatial and temporal style variations, respectively. The foot contact network is trained for the removal of foot skates so that realism in style transfer may be achieved. These three sub-tasks are collaboratively executed in a pipeline to accomplish motion style transfer. A brief overview of Smith et al.’s [8] method for motion transfer is presented in Figure 3.

The proposed method by Smith et al. [8] takes heterogeneous motion sequences from the 3D motion data as input and it outputs the same motion sequence but in the desired target style. The input motion data captured by Xia et al. [25] was used for this purpose. Furthermore, the existing 3D motion dataset was augmented by adding seven more style variations. The poses from the input dataset were annotated with foot contacts and similar poses with different styles were registered together. The pose and timing networks were trained by observing the positions and velocities of the joints and fed as input through window-based pose inputs. Furthermore, a window-based pose input was fed to the foot stake network for the removal of foot staking.

The training phase for the proposed model took 3 h on an i7 3.5 GHz (four-core) GPU-based machine with a GeForce GTX 1070 graphics card. Once the training was complete, the dataset could be discarded. Once the output was achieved from all three networks, the outputs were combined for the target style. The final output was validated in three different ways: (i) by comparing the speed and memory usage with the existing work, (ii) by comparing the visual results, and (iii) through a user study. The comparison of speed and memory footprints shows that the Smith et al. [8] model outperforms the other competitors. The visual results of seven different motion actions are provided and show reasonably better visual results than the other models. The user study for the validation contains two different experiments and concludes that, except for the style, which needs significant variations in limb positions (proud or sexy walk), the model can effectively extrapolate the different styles.

Aberman et al. [11] propose a framework for motion style transfer capable of extracting styles from video clips directly. Furthermore, the framework can extract styles from 3D-animated characters, and 2D projection of 3D motions. The proposed framework trains by unpaired motions with style labels and can transfer the styles from motion, even if the styles are not observed during the training phase. The framework employs a deep convolutional neural network that trains a universal style extractor. This universal style extractor is capable of extracting styles from 3D and 2D motions (video clips). Motion style transfer from video clips proposed by Aberman et al. [11] is shown in Figure 4.

The proposed framework uses two different motion clips as input: a content input with source style and a style motion with the target style. The content motion and style motion are treated differently. Content motions are represented by joint rotations, whereas style motion is presented by joint positions. The output of the framework is the motion content with a target style. Both input motions are encoded into separate latent codes by employing two different encoders. The style encoder further uses 2D or 3D encoders based on the joint coordinates in either 3D or 2D (extracted by video clip) for style extraction. The latent code for style is obtained from only one encoder (either 3D or 2D), depending on the source of the input style.

Considering the dissimilarity of the contents for input motions, the translation of global velocity needs to be performed so that motion style transfer looks realistic. Aberman et al. [11] present a heuristic solution and perform the temporal average for the velocity of the maximal local joint in each motion sequence. For realistic style transfer, foot stacking also needs to be addressed. The proposed framework extracts the foot contact label from the content input and performs inverse kinematics to the corresponding output style.

To implement the style extraction framework, Aberman et al. [11] utilized two different datasets. Firstly, they used the dataset of Xia et al. [25], which contains eight different style labels for various motion sequences. Additionally, they captured a dataset containing several motions performed by a character in 16 different styles. The experiments for validating the proposed framework utilized both of these datasets, with a total of 1500 motion sequences from the first dataset and 10,500 motion sequences from the second dataset. The motion samples were represented as fixed-size windows. The training phase for the first dataset took 8 h, while for the second dataset, the training time was 16 h. Moreover, 10% of the total motion sequences were used as test data for the experiments.

The effectiveness of the proposed framework is validated by comparison with Holden et al.’s approach [9]. Holden et al. extract style from 3D motion; however, Aberman et al. use a state-of-the-art pose estimation algorithm to extract style from videos for a fair comparison. Validation is achieved through a case study in which both approaches are evaluated based on realism, repulsiveness, and content preservation. For the case study, 22 subjects were shown the results and asked for their responses. 132 responses were received, and about 75% of the respondents endorsed Aberman et al.’s results [11].

Xia et al. [25] proposed an online learning method for motion style transfer. The goal of the proposed method is to transfer the input motion data into a series of output-style frames. The method establishes a set of local collections of auto-regressive models and focuses on the deviations in the input and output styles. The regressive models match the identical examples of every input pose from the training data. Afterward, the input pose is extracted from the output style via linear transformations. For every successive pose, a new local model is built and performs the transformation.

The proposed model in [25] is a data-driven method that consists of three major components: (i) motion registration and annotation, (ii) stylistic motion modeling and generation, and (iii) post-processing. In the motion registration and annotation phase, all actions in the input database are registered against motion examples. After the registration process, the motion examples are annotated in terms of actions, styles, and contact information of the motion. The stylistic motion modeling and generation phase uses the input motion database to model the spatial–temporal relationship between the styles. In this phase, an online learning approach is used to automatically build a set of regressive models. These local regressive models help approximate the spatial–temporal transformations and are built by the motion examples corresponding to every input style in the database. The local regressive models can handle the unlabeled input motions. The synthesized motion sometimes violates kinematic constraints enforced by the environment due to environmental contact. Thus, the post-processing phase is performed to automatically annotate such constraints in motion data. An overview of Xia et al. ’s [25] method for motion transfer is presented in Figure 5.

The authors of [25] created a motion dataset by capturing heterogeneous stylistic motions. They captured a wide variety of motion actions and styles to create a comprehensive dataset and then registered structurally identical motions with corresponding styles, annotating all motions, actions, and styles. To make the motions more realistic, post-processing was performed to remove foot stakes. Binary footprint annotations were defined for each pose, and a *K*-nearest neighbors algorithm was used to identify the likelihood of a foot stake artifact. Inverse kinematics was used to identify the contact plane of the contact point.

Xia et al. [25] implemented their model on a machine equipped with an Intel(R) Xeon(R) processor E3-1240 with a speed of 3.40 GHz and an NVIDIA graphics card GTX 780T with 3 GB of memory. The validation of results was achieved by comparing them with existing models. A number of parameters were used for evaluations using different datasets. The comparison results show that the Xia et al. [25] model consistently outperforms its competitors.

Zhang et al. [26] present a neural network-based model named Mode-Adaptive Neural Networks (MANN) to control quadruped characters. MANN is a time series prediction system that predicts the character’s state by learning from its state in the previous frame. The architecture of MANN mainly consists of two components: (i) a motion prediction network and (ii) a gating network. The weights of the motion prediction network are dynamically computed with the help of the gating network. The gating network uses motion features from the input and computes the expert weights for the motion prediction network. MANN architecture for motion style transfer is shown in Figure 6.

The input and output data of MANN include body joint transformations, ground trajectories, and velocities of the character in the previous and following frames, respectively. Specifically, the input contains the trajectory positions, trajectory velocities, joint positions, rotations, velocities, trajectory forward-facing directions, and one-hot vectors for action types relative to the current state. The output contains the predicted values for the character in the next frame.

Zhang et al. [26] created a motion dataset for quadrupeds by capturing 30 min of motion from a dog in different locomotions, including idle lying, standing, walking, sitting, jumping, and trotting. The data was captured on flat terrain and mirrored to increase its size. The network was trained with the captured quadruped motion data so that for a given input feature matrix, the corresponding output can be generated with minimal error. Considering time-series prediction, the training objective was to reduce the mean square error between the input and the predicted value. The training was performed using 4 and 8 expert weights for the prediction networks and took 20 and 30 h on an NVIDIA GeForce GTX 970 GPU, respectively.

Zhang et al. [26] evaluated the proposed model by comparing it with existing neural network-based models using performance metrics such as motion quality, foot sliding artifacts, leg stiffness, and responsiveness. The comparison study concludes that MANN produces better results than the other models.

Mason et al. [27] proposed a transfer learning method that learns neural networks for motion style transfer. The goal of the proposed model is to leverage the already trained models for transforming the new styles. The proposed model limits the amount of required data by using few-shot learning from existing, limited data to generate new motion styles. The model is trained on a few styles using a phase-functioned neural network (PFNN), from which several style-agnostic attributes are extracted. Such style-agnostic attributes along with other particular style attributes combine to produce the output. Few-shot learning for motion style transfer from motion-captured datasets is shown in Figure 7.

The proposed model is required to process style-dependent as well as style-independent components. The proposed model uses PFNN to process motion components that are independent of any style, and several residual adapters to process style-dependent components. The weights of the residual adapters are split into three tensors using a canonical polyadic (CP) decomposition, which aims to limit the attributes needed to learn new styles, enabling few-shot learning. The PFNN and residual adapters are initially trained using a large set of motion data and corresponding styles, with several shot motion clips used for further training.

To implement the proposed approach, Mason et al. [27] captured their own motion dataset analogous to the one captured by Xia et al. [25], and mirrored it to increase the volume of unaligned stylized motions. To achieve an identical skeleton structure (as in the CMU dataset and Holden dataset [7]), velocities, joint positions, and rotations were re-calculated and normalized for every frame. The model was trained on an NVIDIA GTX 1080 Ti GPU, taking 6 h for the training. The generated results were compared to existing work for effectiveness, and the comparison shows that the proposed model of Mason et al. [27] can learn new styles with less training time than other algorithms. Furthermore, the results were validated by visually inspecting the quality of videos generated by different models. The visual evaluation shows that the model by Mason et al. [27] produces better-quality videos than the others.

Dong et al. [28] present a motion style transfer algorithm named adult-to-child (*Audlt2child*). Motion style transfer for children is different from adult motion transfer due to differences in skeleton size, limb dimensions, and movement speeds. Therefore, motion style transfers from adult examples cannot be directly applied to child motion data. Furthermore, motion-capturing for children is not easy, as a child’s level of understanding differs from an adult’s; children lack patience and are confused with instructions.

The goal of the proposed approach is to divide the motion sequences into smaller movements, which can then be translated into images using image translation algorithms. These smaller temporal windows are capable of carrying the temporal and spatial properties of motions, including temporal evolution, and can also be used to capture the essence and stylistic behaviors of those motions. The *audlt2child* motion transfer algorithm proposed by Dong et al. [28] is based on generative adversarial networks (GANs). GANs have the potential to build a mapping between two different data sources. The adult and child have distinct motion attributes and both lack motion alignments. Thus, in this situation, GANs are the preferred choices for the style transfer from the adult to the child. The *audlt2child* uses architecture based on CycleGAN so that timing of motion can be altered through the neural network. The useful attributes from input motions are extracted by redesigning the generators.

Typically, CycleGAN-based architectures employ unpaired data of two domains, such as images of zebras and horses. In *audlt2child* architecture, Dong et al. [28] use the same motion types (such as adult and child jump) and train the network accordingly. The architecture is composed of two separate GANs, one is for *adult2child* motion style extraction and the second one is for *child2adult* motion style translation. Both are executed cyclically. The reason for having two separate GANs with cyclic formation is that this arrangement avoids the need for paired data for training purposes.

Dong et al. [28] built their own motion dataset to address the lack of existing datasets for children’s motions. They captured similar motions for children and adults to populate the dataset with a variety of movements. The model was implemented using Google Colab Pro with an NVIDIA P100 graphics card, and the training phase lasted for 7 h. Dong et al. [28] employed different methods to evaluate the generated results, including comparing them with existing work, conducting an ablation study, and performing a perceptual study. The evaluation of results concluded that Dong et al.’s [28] model generates more realistic results.

Tao et al. [29] proposed a style encoder–recurrent–decoder (ERD) framework for online motion style transfer in real-time. The proposed style-ERD framework considers styles as input streams and processes them online in a streaming fashion. The style-ERD framework generates high-quality motion style transfer in real time by embedding the knowledge of prior frames in the memory of the style transfer module. The proposed framework is based on the ERD model, which consists of several hidden layers forming several recurrent residual connections. These recurrent layers help store the context (content and style) of the current frame, which is used for style extraction.

The style transfer module of the proposed style-ERD framework consists of three components: (i) an encoder, (ii) a recurrent module, and (iii) a decoder. The encoder in the style transfer module compresses the input frame from the motion data, and the recurrent module uses the residual connections for learning motion styles. The decoder maps the latent codes back to the extracted motion frame. Tao et al. [29] use motion capture data as input for their proposed framework and extract styles for the targeted motion. They use the Mocap dataset captured by Xia et al. [25] for this purpose.

The training process of style ERD is performed analogously to the standard generative adversarial network (GAN). Tao et al. [29] evaluated its effectiveness in a variety of ways. Firstly, style transfer results were compared with the existing style transfer models; secondly, qualitative evaluation was performed by considering three parameters, i.e., style expressiveness, temporal consistency, and content preservation; thirdly, a user study was performed to evaluate the quality of results and run-time efficiency of the framework. Tao et al. [29] measured the runtime efficiency with a machine equipped with NVIDIA GeForce GTX 1060 GPU with 6 GB of memory. The evaluation results show that the style ERD framework outperforms the existing models in all evaluation scenarios.

Chang et al. [30] propose a probabilistic model called denoising diffusion probabilistic model (DDPM) for styled motion synthesis. The proposed model is a diffusion-based solution used for combing motion synthesis and style transfer. Furthermore, it models the content and style in a shared representation. The multi-task architecture of the proposed DDPM models various aspects of motion, such as joint angles, foot contact patterns, and global movements. Furthermore, it employs adversarial training for harmonizing the predictions performed by multiple tasks so that the synthesized motions may be perceived globally in a natural way.

The architecture of DDPM uses three inputs (i) a motion clip, (ii) motion content, and (iii) motion style. It models the motion as joint angles and the motion content and motion style are altered into a ’one-hot embedding’. The architecture utilizes the three inputs and estimates the noise, global movement, and corresponding foot contacts. It leverages the binary representation to identify the foot contact with the ground. The multi-tasking architecture of DDPM utilizes noise predictions rather than predicting the joint angles.

Chang et al. [30] utilized Mocap data captured by Xia et al. [25] to provide a variety of motions for qualitative analysis of motion generation with various contents and styles. The authors trained their model using an NVIDIA RTX 3090 GPU with 32-bit floating-point arithmetic, and the training phase lasted for about 24 h. Chang et al. [30] evaluated their model in various ways, such as by comparing it with existing DDPM-based models, qualitatively evaluating results by visualization, and using an ablation study. The evaluation concludes that the designed model generates higher-quality motions than its competitors.

Jiang et al. [31] presented a framework called a motion puzzle that transfers motion styles by body parts. The proposed framework extracts style from a variety of motions for several body parts. The extracted styles are then locally translated to the desired body parts. The motion puzzle framework is designed to preserve the content of the specific motion by controlling the styles of individual body parts (such as the spine, legs, and arms). The proposed framework takes a motion source and several desired motions as inputs. The targeted motions stylize the body parts, and the full-body motion is generated as the output.

Jiang et al. [31] used encoders to design the motion puzzle framework. The design comprises a decoder that creates a styled whole-body motion, and two encoders that extract multi-level style characteristics for individual body parts from target movements, as well as a content feature from a source motion. The authors used a spatial–temporal convolutional network as the foundation for the proposed motion puzzle framework to encode and synthesize motions in which several joints move over time in spatially and temporally correlated ways. The symmetry between the structure of human body parts and the convolutional network is achieved by employing pooling–unpooling methods.

The authors developed their dataset by extracting motion samples from the CMU dataset. The Mocap dataset captured by Xia et al. [25] was used to evaluate the designed model. The model training was performed with two NVIDIA GTX 2080ti, and it took 5 h. The evaluation was performed by (i) visualizing the results for transferring styles by body part, (ii) comparing it with existing models, and (iii) conducting an ablation study. The authors demonstrated the integration of an existing character controller with a real-time motion style transfer.

Motion styles can be transferred from one sequence to another using adversarial learning. Wang et al. [32] proposed a neural network-based architecture for achieving motion style transfer. The architecture consists of a sequential adversarial autoencoder (SAAE) and a style discriminator. At each time step, the SAAE takes a one-hot encoding of the input sequence’s style/emotion label as input. The style discriminator learns to extract the input sequence’s style information from its encoding representation. The model uses motion examples from the Mocap dataset and transfers one motion’s style to another. Wang et al. [32] conducted experiments on their model using the Emilya [33] dataset. The validation of results was performed by comparing them with the existing models. We present the implementation details of various motion style transfer approaches in Table 1.

## 3. Tools and Technologies for Motion Style Transfer

In the existing literature, researchers employ several tools and state-of-the-art technologies to facilitate motion style transfer. Network architecture plays a key role in learning deep representation to facilitate motion style transfer. Network architectures are leveraged to learn spatial and temporal correlations in motion data for efficient style transfer. The architectures for learning deep representations for motion style transfer can be broadly classified into three categories [34]: (i) architectures for learning spatial features, (ii) architectures for learning temporal features, and (iii) miscellaneous architectures. This section provides a brief description of each category.

### 3.1. Architectures for Learning Spatial Features

Spatially-structured architectures facilitate DNNs to learn spatial correlations that rely on network architectures. These architectures are structured around the human skeleton, such that the function of the network computes inherent information about the human skeleton. These approaches process the data either hierarchically or in parallel network branches after dividing the skeleton into body parts. Convolutional neural networks and graph convolutional networks are well-known architectures of this category.

#### 3.1.1. Convolutional Neural Networks

Convolutional neural networks (CNNs) are widely used in image processing, pattern recognition, and motion style transfer. When learning spatial correlations in data with regular structures, such as images and videos, CNNs are particularly efficient. CNNs can be applied to transfer the style of one animation frame to the content of another. CNN is a fully connected network, i.e., all neurons are fully connected with the neurons of other layers. The architecture of CNN is similar to the ANN and it contains many hidden layers in addition to one input layer and one out layer. The hidden layers of CNN may be convolutional, fully connected, and or normalization layers. A linear unit rectifier (ReLU) is generally used as the activation function for CNN. A reverse-propagation function may be adopted for the weight assignments between the successive layers of CNN. Pan et al. [1] and Aberman et al. [11] employ CNN for motion style transfers.

#### 3.1.2. Graph Convolutional Networks

Graph convolutional networks (GCNs) are extended versions of CNNs that have emerged recently for skeleton motion. The key difference between CNN and GCN is that the former is typically executed on regular and structured data, while the latter is a more generalized version that can operate on irregular data. The GCN constructs a map of nodes and links the neighborhood nodes via Euclidean distances on which a convolution is applied. The GCN learns the features by considering the Euclidean distances between the neighboring nodes. It uses the feature vector for sampling and execution in the Fourier domain. Aberman et al. [35] and Li et al. [36] used GCNs to design the models for motion generation.

### 3.2. Architectures for Learning Temporal Features

The temporal dimensions of motion data provide data on the types of activity performed and how they are carried out. We provide popular architectures in this sub-section.

#### 3.2.1. Recurrent Neural Networks

Recurrent neural networks (RNNs) are specialized forms of neural networks and are widely used for sequential data, such as sound, images, etc. RNNs typically execute timestamps of time series data sequentially and can handle different lengths of sequences. Such networks build internal states to capture the temporal context. RNNs use the hidden layer to retain sequential information in memory, enabling them to store time-based sequences. They can be used to predict subsequent poses in human motion. In their work, Jain et al. [37] used one RNN for each body part and predicted the next poses by incorporating the predictions of neighboring RNNs (at the previous timestamp) and its own prior predictions as inputs. RNNs are well-suited for learning time series data like human motion, as they can predict the next posture from the previous motion. However, they are known to converge to average poses.

#### 3.2.2. Long Short-Term Memory

Generally, the RNN uses long short-term memory (LSTM) architecture for learning temporal features. A typical LSTM contains a memory cell that may retain values for any length of time and three gates that control the flow of data into and out of the cell. Because the memory cell may retain data over any time, LSTM is useful for capturing both short-term and long-term temporal relationships. These gates are an input gate, an output gate, and a forget gate. The gradients used to update the network weights might become large (or extremely small), which either precludes the further learning of the network or causes the network to diverge. Wang et al. [32] used LSTM-based learning for style transfer.

### 3.3. Miscellaneous Architectures

Several authors [38,39,40] have designed DL architectures to learn special–temporal relationships for motion style transfer. These architectures aim to model the temporal motion parameters and decompose the spectral motion parameters for spatiotemporal style transfer [41]. For example, Zhang et al. [26] made adjustments in expert weights by using a gating network; this resulted in a mixture of the scheme.

## 4. Datasets for Skeleton Motion

This section provides an overview of existing popular datasets for skeleton motion. The dataset selection is an important aspect when designing an algorithm. The designed model is trained from the dataset and the quality and comprehensiveness of the dataset directly influence the performance of the model. Therefore, it is crucial to choose an appropriate dataset for the model. This section provides an overview of existing popular and widely used motion datasets. This overview of existing datasets will be very beneficial for researchers to choose the appropriate dataset.

### 4.1. CMU Dataset

The Carnegie Mellon University (CMU) motion capture database is one of the most widely used datasets. The CMU dataset is a large-scale freely available dataset for learning and research purposes. The CMU dataset contains about $3.9\times 10^{6}$ frames recorded at a frequency of 120 Hz. The movements of 29 joints are tracked for the motion data. The dataset contains motions of various types organized into 144 classes and 2605 motion samples. The data are arranged into 6 major categories—human interaction, interaction with the environment, locomotion, physical activities, sports, situations, scenarios, and test motions. The data are available in several formats, such as ASF, AMC, and zip (compressed), with the possibility of converting to several other formats. The total size of the data is over 4 GB.

### 4.2. Human3.6M Dataset

*Human3.6M* [42] is a diverse and large-scale dataset of human poses. It is a collection of 3.6 million 3D human poses. The dataset was generated by recording the movements and poses of 6 male and 5 female professional actors. The movements and poses in 15 different scenarios were recorded and the dataset was organized accordingly; 32 joints of the skeletons were recorded for the movements. The *Human3.6M* dataset is not freely available; however, it can be obtained upon request. Moreover, 941 citations of the *Human3.6M* dataset demonstrate its wide popularity in the research community.

### 4.3. Mocap HDM05 Dataset

The Mocap HDM05 [43] is a publicly available large-scale motion dataset containing over 360,000 frames recorded at a frequency of 120 Hz. The dataset is available in various formats, such as C3D, ASF, AVI, and AMC, and includes 31 marked joints of the skeleton, with more than 3 h of systematically recorded motions organized into over 100 different classes. The dataset was populated by five actors performing various motions between 10 to 50 times. The HDM05 [43] is widely accepted among the research community and has over 462 citations.

### 4.4. NTU RGB+D Action Recognition Dataset

Shahroudy et al. [44] presented the NTU RGB+D dataset, which is a large-scale 3D dataset for human activities. The dataset was generated using Microsoft Kinect v2 sensors to track 25 joints in the 3-dimensional human body. It contains 56,000 videos and 4 million frames with a frequency of 30 Hz, organized into 60 action classes further divided into 3 major groups: 40 daily routine activities (reading, eating, etc.), 9 health-related actions (falling, sneezing, etc.), and 11 other hybrid actions (hugging, kicking, etc.). The RGB+D dataset is widely adopted by the research community and has been cited over 1680 times.

### 4.5. NTU RGB+D 120 Action Recognition Dataset

Lie et al. [45] introduced RGB+D 120, a large-scale dataset for human activity recognition. RGB+D 120 is an extended form of the RGB+D dataset, which is based on similar sensors and methodologies; however, the scale of the dataset is much higher. The RGB+D 120 dataset was created by tracking 25 human body joints. The dataset captures 120 action categories of 40 persons; the categories include 82 daily routine activities, 12 health-related actions, and 26 other hybrid action categories. The dataset contains 114,000 videos and 8 million frames at a frequency of 30 Hz. The NTU RGB+D 120 dataset has 427 citations.

### 4.6. HumanEva Datasets

Sigal et al. [46] presented *HumanEva*, a video and motion capture dataset. Specifically, Sigal et al. [46] developed two datasets and named them *HumanEva-I* and *HumanEva-II*, respectively. The *HumanEva-I* dataset was developed by four subjects who exercise six predefined actions repeatedly. The *HumanEva-I* dataset comprises 74,600 frames recorded at a 60 Hz frequency. *HumanEva-II* is a relatively smaller dataset, consisting of 24,600 frames at a 60 Hz frequency. Both datasets track the motions of the human body by monitoring 15 different body joints. The citation count for the *HumanEva* datasets is 1073.

### 4.7. Holden et al. Motion Dataset

Holden et al. constructed a motion dataset [9] by using existing datasets, such as CMU and HDM05 by [43] and Xia et al. [25]. After combining these existing datasets, Holden et al. captured their own data and append them to the existing datasets. In order to ensure symmetry between the skeleton structures of different datasets, retargeting operations utilizing inverse kinematics are carried out. The resulting dataset, as compiled by Holden et al., comprises 6 million frames captured by tracking 21 body joints at a frequency of 120 Hz. The Holden et al. constructed dataset [9] is widely adopted by the research community; its popularity is evident from 462 citations.

### 4.8. 3DPW Dataset

Marcard et al. [47] presented *3D poses in the wild* (3DPW) dataset, which records videos from a moving phone camera. The 3DPW dataset contains 60 video sequences with a total of 51,000 frames recorded at a frequency of 30 HZ. The dataset contains 11,000 different motions captured by tracking 52 body joints. The dataset was recorded with 18 3D models in different arrangements. The dataset is freely available for research activities. The 3DPW dataset has 392 citations.

### 4.9. AMASS Dataset

Mahmood et al. [48] presented the archive of motion capture as surface shapes (*AMASS*) dataset, which consists of large-scale data on human motions. AMASS is a publicly available dataset used for research purposes and consists of 440 different subjects and 13,195 motions. The dataset contains 180 million frames with a frequency of 60 Hz. The citation count for the *AMASS* dataset is 282.

### 4.10. Total Capture Dataset

Trumble et al. [49] presented a human pose dataset named *total capture* that is publicly available. The dataset was built by using the multi-viewpoint video (MVV) and inertial measurement unit (IMU). The dataset was created by 5 actors (4 males and 1 female) performing different actions in various poses with 3 repetitions. The dataset contains 1,892,176 frames recorded at a frequency of 60 Hz. The citation count for the *total capture* dataset is 133. Summary of widely used motion datasets is given in Table 2.

## 5. Current Challenges and Future Research Directions

This section briefly describes the contemporary challenges of motion style transfer techniques. Despite considerable research efforts for motion style transfer, there are still several challenges that exist that preclude the popularity of such techniques. A few prominent research challenges in the domain are presented as follows.

### 5.1. Quantitative Evaluation of the Synthesized/Generated Motion

Anticipating the significance of the domain, more models are being developed to facilitate motion style transfer. However, to the best of our knowledge, there is no unified model that quantitatively evaluates the generated motions. Generally, the researchers are evaluating the generated motions by visual representation, case studies, and/or comparing them with previous work. There is a lack of literature on the quantitative evaluation of synthesized/generated motions. The quantitative evaluation of synthesized/generated motions is an open research challenge.

### 5.2. Non-Availability of Benchmark for Evaluation

Motion style transfer has been an active area of research for over a decade, with several researchers utilizing various enabling technologies, hardware, and datasets, and with varying training times (Table 1). For an effective evaluation of the designed approach, a unified benchmark needs to be developed that takes into account several parameters, such as hardware specifications, datasets, training time, and unified performance metrics, in order to assess the effectiveness of these techniques.

### 5.3. Style Retargeting

When extracting a motion style from a source to be transferred to the destination character, it must be consistent with the skeleton structure of the destination character. In case of a mismatch between the skeleton structure of the source and destination, style retargeting needs to be performed. In style retargeting, the extracted style is tailored such that it may be applied to the destination skeleton.

### 5.4. Need for Paired and Registered Data

The source of motion examples is of key importance for efficient motion style transfer, as the styles are extracted from the source motion. These motion sources need to be comprehensive and may contain diverse motions and style pairs. Motion style transfer techniques use motion data in which different styles of motion can be paired and registered together [11].

To build such data, each character needs to perform specific actions in different styles, such as performing a walk in several different styles, and all such styles need to be paired and registered together. However, this is an extremely time-consuming and non-scalable approach. Therefore, for motion style transfer, it is necessary to have paired and registered data, which is a challenging and difficult task to achieve.

### 5.5. Style Extraction and Transfer from Fewer Examples

Due to the difficulty of capturing and processing sufficient motion style data, motion style transfer techniques should be capable of extracting and transferring styles from fewer motion examples.

### 5.6. Training Overheads

Anticipating the insufficiency of traditional motion style transfer techniques, the researchers employed various AI-based approaches to efficiently transfer the motion styles. These AI-based approaches train the model and perform motion style transfer subsequently. The AI-based techniques have shown great potential and are successfully applied for motion style transfer. The training overheads of such an approach affect their performances. The availability of comprehensive training data and the overheads for training the models are significant challenges associated with AI-based motion style transfer approaches.

### 5.7. Handling Unseen Motions/Styles

To transfer the motion styles, the model needs to cope with the unseen motions/styles. Generally, motion style transfer performs retraining to address unseen motions and styles. This retraining consumes further computing resources and, hence, degrades the performance of the style transfer technique. Efficiently handling unseen motions/styles without performance degradation is challenging for motion style transfer techniques.

## 6. Conclusions

In today’s dynamic and interactive digital world, the utilization and modeling of video characters for animations have increased significantly. Motion style transfer is an active approach for modeling human-based characters for use in animation applications. This paper aims to provide a comprehensive and concise overview of motion style transfer techniques. It presents a brief overview of state-of-the-art motion style transfer techniques, along with their enabling technologies and implementation details. The sources of motion examples for style transfer play crucial roles; therefore, an overview of several prominent datasets is also provided in this study. Finally, the existing challenges in the area are highlighted.

## Figures and Tables

**Figure 1 sensors-23-02597-f001:**
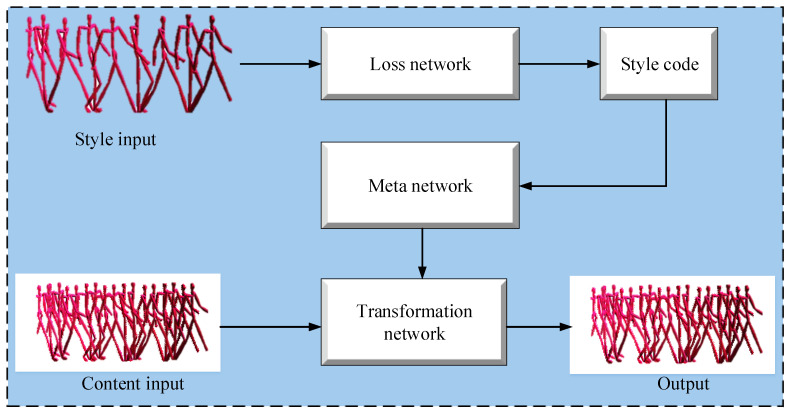
Meta network-based motion style transfer from motion-captured datasets.

**Figure 2 sensors-23-02597-f002:**
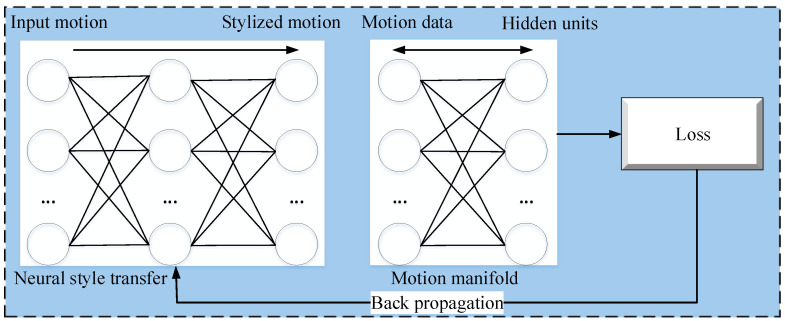
A typical architecture of style transfer from motion-captured datasets.

**Figure 3 sensors-23-02597-f003:**
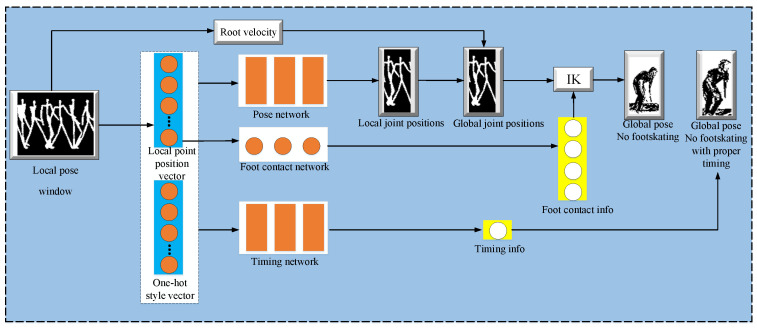
Real-time motion style transfer from motion-captured datasets.

**Figure 4 sensors-23-02597-f004:**
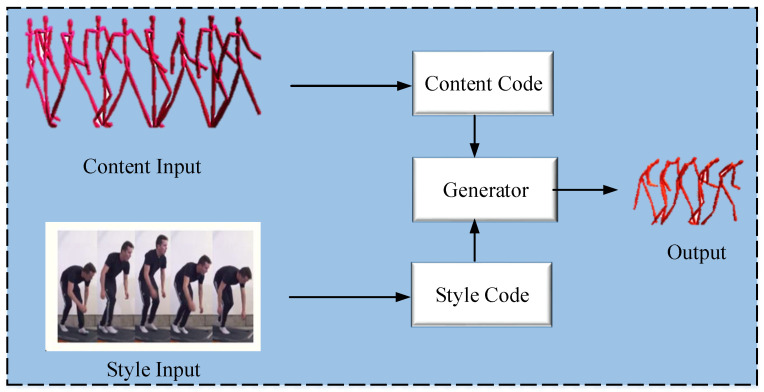
Motion style transfer from video clips.

**Figure 5 sensors-23-02597-f005:**
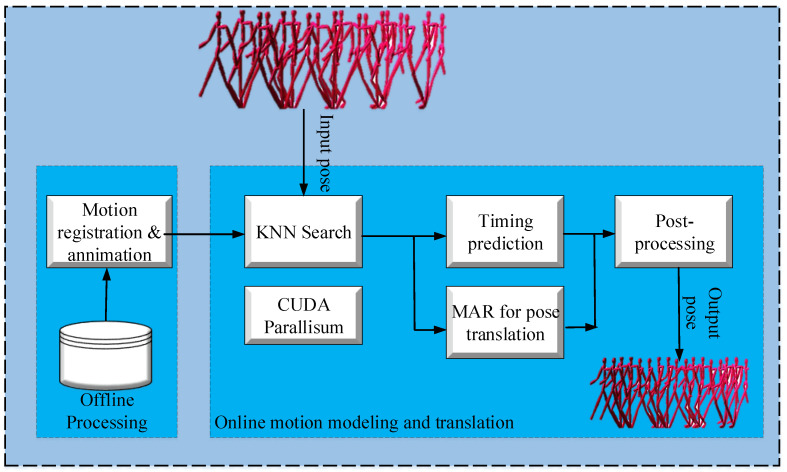
Regression-based motion style transfer from motion-captured datasets.

**Figure 6 sensors-23-02597-f006:**
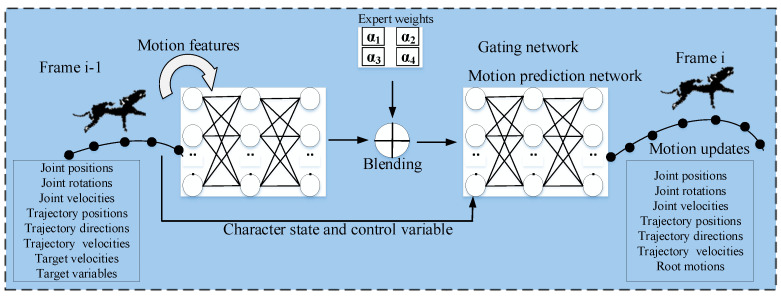
An architecture of MANN for motion style transfer.

**Figure 7 sensors-23-02597-f007:**
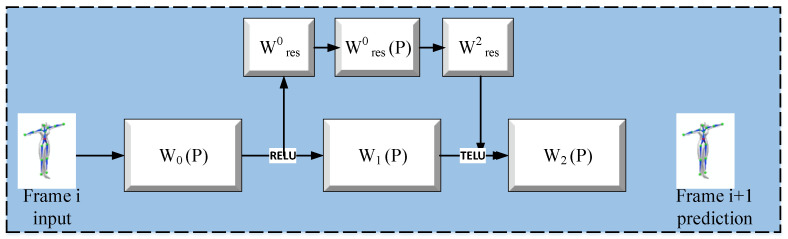
Transfer learning for motion style transfer from motion-captured datasets.

**Table 1 sensors-23-02597-t001:** Implementation details of motion style transfer approaches.

Authors	Enabling Technology	Datasets	Hardware	Training Time	Result Evaluation
Pan et al. [1]	CNN	Uses different datasets Dataset 1: MHAD, with 660 action sequences Dataset 2: CMU motion capture dataset Dataset 3: HDM05 Dataset 4: Holden et al. [9] **Motion actions/Styles:** Details of datasets (2–4) is provided in Section 4	NVIDIA GeForce GTX1050Ti,	10 h	Result Visualization Comparison with existing work Style interpolation
${}^{*}$ Holden et al. [7]	CNN	Holden et al. [9] dataset with 10 motion styles, details of datasets is provided in Section 4. **Motion actions/Styles:** 10 different styles and various kinds of locomotion including walking, jogging, and running	NVIDIA GeForce GTX 970	3.5 h	Comparison with existing work
Smith et al. [8]	ANN	Uses two different datasets Dataset 1: 79,000 motion sequences (captured by Xia et al. [25]) Dataset 2: 550,000 motion sequences (self-captured) **Motion actions/Styles:** walking, running, jumping, kicking, punching, neutral, proud, angry, depressed, strutting, childlike, old, sexy	i7 3.50 GHz, 4 core PC with a GeForce GTX 1070 graphics card	3 h	Result Visualization Comparison with existing work Case study
${}^{*}$ Aberman et al. [11]	GNN/encoders	Uses two different datasets Dataset 1: 1500 labeled motion sequences (captured by Xia et al. [25]) Dataset 2: 10,500 total motion sequences (self-captured) **Motion actions/Styles:** walking, running, jumping, kicking, punching, neutral, proud, angry, depressed, strutting, childlike, old, sexy	NVIDIA GeForce GTX Titan Xp GPU (12 GB)	8 h and 16 h for dataset 1 and dataset 2, respectively	Result Visualization Comparison with existing work Ablation Study Style Interpolation
Xia et al. [25]	MAR	Uses two different datasets Dataset 1: CMU motion capture dataset (details of dataset is provided in Section 4) Dataset 2: 79,829 motion sequences (self-captured) **Motion Actions/Styles:** As of Smith et al. [8]	Intel(R) Xeon(R) E3-1240 3.40 GHz and NVIDIA GTX 780T (3 GB)	Unknown	Comparison with existing work Component Evaluations
${}^{*}$ Zhang et al. [26]	ANN	Synthetically created 30 min dog motion capture dataset **Motion actions/styles:** walk, pace, trot, and canter, sitting, standing, idling, lying and jumping	NVIDIA GeForce GTX 970 GPU	20/30 h with 4/8 expert weights	Comparison with existing work
${}^{*}$ Mason et al. [27]	ANN	Synthetically created 1 h motion capture dataset **Motion actions/Styles:** walking, running, jumping, kicking, punching, neutral, proud, angry, depressed, strutting, childlike, old, sexy	Intel i7-6700HQ CPU	7 h	Result Visualization Comparison with existing work
Dong et al. [28]	CycleGAN	Synthetically created dataset that contains 23 motion actions performed by both adults and children. **Motion actions/Styles:** Ball throw with left /right arm, punch, kick, jump, idle, broad jump forward, high jump, 5 jumping jacks, walk, fast walk, hopscotch, sneaky walk, happy walk, jog, fast run, skip	NVIDIA P100 graphics card	7 h	Comparison with existing work Ablation study Perceptual study
${}^{*}$ Style ERD [29]	GNN	Mocap dataset captured by Xia et al. [25] **Motion Actions/Styles:** As of Xia et al. [25]	PC with NVIDIA GeForce GTX 1060 GPU (6 GB)	Unknown	Comparison with existing work Qualitative evaluation user study
${}^{*}$ Chang et al. [30]	DDPM	Mocap dataset captured by Xia et al. [25] **Motion actions/styles:** As of Xia et al. [25]	NVIDIA RTX 3090 GPU, 32 bits	24 h	Comparison with existing work Qualitative evaluation Ablation study
${}^{*}$ Motion Puzzle [31]	CNN	(i) CMU dataset, and (ii) Mocap dataset captured by Xia et al. [25]	2x NVIDIA GTX 2080ti	5 h	Results visualizing Comparison with existing work Ablation study

* Source code publicly available.

**Table 2 sensors-23-02597-t002:** Summary of motion datasets.

Dataset	Size	Frame Rate	Joints	Citations	Availability	URL
**CMU Dataset** Motions/actions: various motion in styles of human interaction, interaction with environment, locomotion, physical activities, and sports scenarios	$3.9\times 10^{6}$ frames	120	29	–	Public	http://Mocap.cs.cmu.edu/, accessed on 12 December 2022
**Human3.6M** Motions/actions: conversations, eating, greeting, talking on phone, posing, sitting, smoking, taking photos, walking in various scenarios	$3.6\times 10^{6}$ frames	120	31	941	On request	http://vision.imar.ro/human3.6m/description.php, accessed on 12 December 2022
**Mocap HDM05 Dataset** Motions/actions: walking, running, jumping, grabbing/depositing activities, sports activities, sitting and lying down, and miscellaneous motions	$3.6\times 10^{5}$ frames	120	31	462	Public	https://resources.mpi-inf.mpg.de/HDM05/, accessed on 20 November 2022
**NTU RGB+D Dataset** Motions/actions: drinking water, eating meal, brushing teeth, brushing hair, sit down, stand up, and several other actions	$4.0\times 10^{5}$ frames	30	25	1680	On request	https://rose1.ntu.edu.sg/dataset/actionRecognition/, accessed on 20 November 2022
**NTU RGB+D 120 Dataset** Motions/actions: put on/take off headphone, bounce ball, tennis bat swing, thumb up/down, make OK/victory sign, and several other actions	$8.0\times 10^{5}$ frames	30	25	427	On request	https://rose1.ntu.edu.sg/dataset/actionRecognition/, accessed on 20 November 2022
**HumanEva-I Dataset** Motions/actions: walking, jogging, gesturing combo, throwing and catching a ball, boxing, combo	74,600 frames	60	15	1137	public	http://humaneva.is.tue.mpg.de/datasets_human_1/, accessed on 5 October 2022
**HumanEva-II Dataset** Motions/actions: walking, jogging, gesturing combo, throwing and catching a ball, boxing, combo	24,600 frames	60	15	1137	Public	http://humaneva.is.tue.mpg.de/datasets_human_2, accessed on 5 October 2022
**Holden et al. motion Dataset** Motions/actions: various motion in styles of human interaction, interaction with environment, locomotion, physical activities, and sports scenarios	$6.0\times 10^{6}$ frames	120	21	462	Public	https://theorangeduck.com/page/deep-learning-framework-character-motion-synthesis-and-editing, accessed on 7 November 2022
**3DPW Dataset** Motions/actions: shopping, doing sports, hugging, discussing, capturing selfies, riding bus, playing guitar, relaxing	$180\times 10^{6}$ frames	60	52	460	Public	https://virtualhumans.mpi-inf.mpg.de/3DPW/, accessed on 7 November 2022
**AMASS Dataset** Motions/actions: walking, jogging, crawling, standing, siting, kicking, ball catching, and many other actions	$5.1\times 10^{4}$ frames	30	23	387	Public	https://amass.is.tue.mpg.de/, accessed on 1 November 2022
**Total Capture Datasets** Motions/actions: walking, running, standing, jumping, kicking, punching, neutral	1,892,176 frames	60	21	160	Public	https://cvssp.org/data/totalcapture/, accessed on 1 November 2022

## Data Availability

Not applicable.

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
