# Peer review of "Deep Learning-Based Motion Style Transfer Tools, Techniques and Future Challenges"

_sensors, 2023, doi:10.3390/s23052597_

Round 1

Author Response

Response to Reviewer 1

Motion style transfer aims to map existing human motion sequences to various styles, which play an important role in animation, film production, and digital entertainment. The author makes a detailed analysis of the related papers on motion style transfer. However, I think this paper still needs to be further improved.

Comment 1
There are some related literature authors who haven't mentioned it. For example, recent motion style transfer works should be reviewed, such as:
(1) Tianxin Tao et al. “Style-ERD: Responsive and Coherent Online Motion Style Transfer” (2023).
(2) Ziyi Chang et al. “Unifying Human Motion Synthesis and Style Transfer with Denoising Diffusion Probabilistic Models” (2022).
(3) Deok-Kyeong Jang et al. “Motion Puzzle: Arbitrary Motion Style Transfer by Body Part” (2023).
By the way, more related works should be mentioned, such as:
(1) Qi Wang et al. “Transferring Style in Motion Capture Sequences with Adversarial Learning” the european symposium on artificial neural networks (2018). Therefore, there are still some works about motion style transfer that the author has not mentioned.

Response:

Thank you for drawing our attention towards some recent and related research articles to include for improving the quality of our manuscript. We have included all the suggested papers. The relevant text is marked with blue color in the revised paper.

 Comment 2
This manuscript aims to provide a survey of “AI-based Motion Style Transfer”. AI, obviously, means artificial intelligence. But, authors provide a survey of “Deep-learning-based Motion Style Transfer”. Data-driven motion style transfer approaches also can extract motion style from existing motion data and implement motion style transfer, but they use other machine learning methods, such as PCA and sparse coding. The authors can refer this two works:
1 Motion Style Extraction Based on Sparse Coding Decomposition. (2019)
2 Learning and transferring motion style using Sparse PCA. (2018)
Hence, if want to summarize AI-based motion style transfer methods, the authors should make a more reasonable and comprehensive classification. By the way, some traditional methods that are not data-driven should also be discussed.

 Response:

Indeed, the manuscript was aimed to provide the overview of AI-based motion style transfer approaches. The initial thought was to mention AI-based motion style transfer approaches by considering the AI and DL as interrelated and AI may cover the DL at macro level. However, we acknowledge the reviewers observation and revise the paper as DL-based motion style transfer approaches, as majority of the approaches mentioned in the paper are DL-based. This makes the paper more specific and comprehensive. The revised text is marked with blue color in the revised version.

Comment 3

In section 3, tools and technologies for motion style transfer are not clearly classified. First, there is no need to introduce the basic knowledge of RNN and CNN in this review. Second, there are some misinterpretations for ANN and DNN. DNN is a type of ANN, and their differences do not exist in the number of hidden layers. Third, due to lack of paired training data, most approaches for motion style transfer are GAN-based. The works of Holden et al. and Aberman et al are both GAN-based. So, I think the authors should make major changes to the third section.

 Response:

The section 3 is thoroughly revised to include the related enabling technologies related to DL. The revised text is marked with blue color in the revised version.

Reviewer 2 Report

This review paper is well structured and listed the details of appoarches between different papers.

One Minor things:

1. ANN suppose includes with CNN. Maybe it is better to define them more clearlier.

 I think it is also good to put infomation of source code access avaliablity,  frameworks and other information relate to deployment which can help researcher to have quick choice to test with these technology.

Author Response

This review paper is well structured and listed the details of appoarches between different papers.

One Minor things:

Comment 1

ANN suppose includes with CNN. Maybe it is better to define them more clearlier.

 Response:

The section 3 is thoroughly revised to clearly address the related enabling technologies. The revised text is highlighted with blue color in the revised version.

Comment 2

I think it is also good to put infomation of source code access avaliablity,  frameworks and other information relate to deployment which can help researcher to have quick choice to test with these technology.

 Response:

The information of availability of source code is mentioned in Table 1. Further the availability of datasets is also mentioned in Table 2. The revised text is marked with red color in the revised version.

Round 2

Author Response

Revision Report based on Reviewer’ Comments

We would like to thank the reviewer for valuable comments which certainly help to improve the readability of the paper. We have incorporated all the comments as detailed below.

_______________________________________________________________________

 Responses to Reviewer’s Comments:

 Point 1: I think the authors still do not give a clear classification for motion style transfer. First. CNN and RNN are both deep neural networks. I cannot understand why authors think deep neural networks, CNN, RNN are different methods for motion style transfer. Second, “3.1 Neural Networks” and “3.2 Feed-Forward Transformation Networks”, I think “Feed-Forward Transformation Networks” are still a kind of neural networks. 3.1 and 3.2 are really two different types?

Response 1: Originally, the paper was intended to provide an overview of the different DNN variants for motion transfer. However, we acknowledge the insight and understand the reviewer's concern that we should classify appropriately. Accordingly, we have rewritten and reorganized Section 3 of the revised version by consulting the previously published literature and classifying the existing architectures accordingly [1]. The existing architectures are classified into three categories: (i) Architectures for learning spatial features, (ii) Architectures for learning temporal features, and (iii) miscellaneous architectures. The revised text is written in blue color. [Please refer to Section 3 of the revised version].

Point 2: The authors said that considering the difficulty in building paired and labeled motion data for style transfer, DNNs are the preferred choice to cope with such issues.” However, it is obvious that most of the deep neural networks are not designed for unsupervised problems such as motion style transfer and style retargeting. Actually, GAN is a kind of network architecture designed for unsupervised problems.

Response 2: We have thoroughly revised Section 3 to make clear the description of different architectures of motion style transfer. The revised text is shown in blue. [Please refer to Section 3 of the revised version].

Overall, I think authors did not clearly understand methods of motion style transfer clearly. I hope authors can give a correct classification of motion style transfer in the revised manuscript.

Response: For the classification, we consulted the existing literature and revised the paper accordingly. The revised classification is analogous to the existing work and cites the same [1]. We hope that the revised classification is acceptable.

[1] Mourot, L.; Hoyet, L.; Le Clerc, F.; Schnitzler, F.; Hellier, P. A Survey on Deep Learning for Skeleton-Based Human Animation. Computer Graphics Forum, 2022, Vol. 41, pp. 122–157

_____________________________________________________________________________________

We have tried our best to improve the manuscript and have made recommended changes in the manuscript. These changes do not affect the scope of the paper. The specific position of the changes in the revised article is explained in detail in this blue-letter response document.

We sincerely thank the editor and reviewers for their work and hope that the corrections will meet with approval.

Once again, thank you very much for your comments and suggestions.
